# Contrast-Enhanced Tissue Processing of Fibrillin-Rich Elastic Fibres for 3D Visualization by Volume Scanning Electron Microscopy

**DOI:** 10.3390/mps4030056

**Published:** 2021-08-15

**Authors:** Philip N. Lewis, Robert D. Young, R. B. Souza, Andrew J. Quantock, Keith M. Meek

**Affiliations:** 1Structural Biophysics Research Group, School of Optometry & Vision Sciences, Cardiff University, Cardiff CF24 4HQ, UK; youngrd@cardiff.ac.uk (R.D.Y.); QuantockAJ@cardiff.ac.uk (A.J.Q.); meekkm@cardiff.ac.uk (K.M.M.); 2Department of Genetics and Evolutionary Biology, University of São Paulo, Rua de Matão 05508-090, Brazil; souza.rb@alumni.usp.br

**Keywords:** fibrillin, elastic fibre, 3D ultrastructure, volume electron microscopy, connective tissue

## Abstract

Elastic fibres constitute an important component of the extracellular matrix and currently are the subject of intensive study in order to elucidate their assembly, function and involvement in cell–matrix interactions and disease. However, few studies to date have investigated the 3D architecture of the elastic fibre system in bulk tissue. We describe a protocol for preparation of tissue samples, including primary fixation and backscatter electron contrast-enhancement steps, through dehydration into stable resin-embedded blocks for volume electron microscopy. The use of low molecular weight tannic acid and alcoholic lead staining are critical stages in this procedure. Block preparation by ultramicrotomy and evaporative metal coating prior to microscopical examination are also described. We present images acquired from serial block face scanning electron microscopy of cornea and aorta showing target structures clearly differentiated from cells and other matrix components. The processing method imparts high contrast to fibrillin-containing elastic fibres, thus facilitating their segmentation and rendering into 3D reconstructions by image analysis software from large serial image datasets.

## 1. Introduction

Besides the main collagenous, structural component of connective tissues, a population of elastic fibres is present, consisting of fibrillin-rich microfibrils around a central core of elastin, which contributes elasticity and resilience to the overall biomechanical properties of the tissue [1,2,3]. Prior to the discovery of fibrillin by Sakai, Keene and Engvall in 1986 [4], a hierarchy of elastic-related structures had been considered to exist in tissues, including oxytalan and elaunin fibres, based on the relative amounts of fibrillar and amorphous component present [5].

It is now recognised that microfibrils with a typical 10–12 nm diameter tubular structure are found in a wide variety of tissues throughout the body, often in association with amorphous elastin, but also independently, as in the lens zonules and cornea [6,7], where they fulfil an important role in tissue extensibility and elasticity. Recent studies of elastic fibre organisation and composition have revealed increased complexity with multiple glycoproteins, notably the fibulin family of proteins, in addition to microfibrils, associated with the fibre surface [8]. Currently, there is intense interest in the mechanisms of fibrillin supramolecular organisation and elastic microfibril assembly [9,10], particularly in relation to their primary structural function, but also to permit a fuller understanding of their presumptive roles in cell–cell and cell–matrix interactions and association with pathologies, such as Marfan syndrome [11] and Cutis laxa [8].

Until quite recently, however, little has been known of the architecture and arrangement of the elastic fibre system in tissues on the mesoscopic scale within relatively large tissue volumes. The development of volume electron microscopy (vEM) techniques, such as serial block face (SBF SEM) and focused ion beam scanning electron microscopy (FIB SEM), has overcome this by enabling serial image acquisition at high resolutions [12,13]. The region of interest at the surface of a resin-embedded tissue specimen can be imaged, then renewed precisely by microtomy or milling, to create a new imaging surface, and the process repeated automatedly to collect many serial images for use in reconstruction of a 3D rendered view. We, and others [7,14,15,16], have used vEM to examine elastic fibre distribution in the cornea, where the presence of elastin and elastic microfibrillar structures has historically been controversial. The imaging technique employs detection of backscattered electrons that are emitted at highest yield by interaction of the microscope electron beam with elements with high atomic number within the specimen; optimal specimen preparation thus aims to incorporate high amounts of metals, such as osmium, uranium and lead, into components of interest within the sample. Additionally, it is beneficial if these high-yield contrasting agents can be attached differentially with respect to different components of the sample, for example, collagen and cells, as this greatly facilitates discrimination between the two when image analysis software is applied later, to generate a 3D model. In essence, a tissue processing method is required which can provide optimal heavy metal contrasting of the elastic microfibril system and, in this protocol, we describe in detail a technique successfully employed for this purpose in our laboratory.

Traditionally, histological stains, such as aldehyde fuchsin, were used to identify by light microscopy microfibrillar elastic structures which, unlike amorphous elastin, were stained after exposure of tissue sections to oxidising reagents. However, such stains generate no backscatter signal on electron bombardment and are thus ineffective for imaging microfibrils by electron microscopy. vEM studies of various tissue systems to date have used a range of tissue-processing schedules, mostly based on an extended method described by Deerinck et al. [17], incorporating double treatment with osmium solutions, followed by uranyl acetate and an *en bloc* lead salt solution. The common aim of current methods in widespread use has been to optimise imaging through backscatter electron emission from all components of the specimen in a non-specific fashion, and there seems to have been few attempts to adjust contrast-processing to optimise and enhance targeted structures, other than through the use of genetic tags [18], which are more readily applied to studies of isolated cells than bulk tissue. Here, we present a protocol for preparation of samples for SBF SEM with specific conditions that facilitate imaging of elastic microfibrils in bulk connective tissue specimens. The method combines a sequence of reagents used more routinely for contrasting ultrathin sections for 2D transmission electron microscopical examination, but with attention to certain key requirements. Small tissue samples are first preserved by conventional aldehyde fixation, then contrasted in standard aqueous osmium tetroxide solution, prior to immersion in tannic acid solution. The use of tannic acid, following osmium, has long been recognised as an effective enhancer of contrast for electron microscopy, particularly for cell membranes, but also for elastin, and is considered to act as a mordant, promoting binding by lead salts applied subsequently [19]. However, problems with uneven penetration into intact tissues has been reported, leading to its use as an on-section stain combined with uranyl acetate. Used in this way, it was shown to provide intense contrast and was suggested to be specific for amorphous elastin [20]. In the method described for tissue contrasting for SBF SEM, we find that low molecular weight (di-Gallic) C_14_H_10_O_9_ tannic acid is critical for full tissue permeation and successful contrasting of elastic microfibrils. The mordanting property of tannic acid seems valuable for effective binding of subsequent reagents, including uranyl acetate and lead salts. Both aqueous and ethanolic uranyl solutions are employed in our protocol. Finally, before resin infiltration and embedding, a novel contrasting step is applied using alcoholic lead acetate solution [21]. Normally recommended for staining of ultrathin resin sections because of its resistance to precipitation of lead carbonate on exposure to air, it is applied here en bloc, which we believe also minimises unwanted lead contamination within the specimen.

## 2. Experimental Design

The protocol would most readily be carried out in a laboratory set up for electron microscopy with expertise in sample processing. This would also provide an environment where risk assessments and standard operating procedures would be in place to allow for the safe handling and disposal of all chemicals used.

The method is based on an adaptation of staining techniques for elastic fibres originally developed for conventional 2D transmission electron microscopy and follows most established preparatory sample procedures, including fixation, staining, epoxy resin embedding, polymerisation, and ultramicrotomy. A fume hood is essential to carry out all staining and resin embedding steps as the chemicals used are toxic when inhaled and should not be allowed to contact skin. Solvent-resistant NBR 92-600 gloves are advised while working with the hazardous chemicals and solvents. Laboratory coats should be worn for added protection from chemicals.

The protocol requires 5 days to fix, stain, infiltrate with resin and embed specimens in moulds.

2 days to polymerize blocks in an oven at 60 °C

3 days for 3view pin preparation

1 day for setting up the SBF SEM microscope for imaging.

3 days to acquire 1000 SBF SEM serial images, at 4 nm/pixel, 4 K × 4 K resolution.

### 2.1. Materials

25% EM Grade Glutaraldehyde (Agar Scientific, Stansted, UK; Catalogue No: AGR1010)Paraformaldehyde (Agar Scientific, Stansted, UK; Catalogue No: AGR1018)Di-sodium hydrogen orthophosphate dihydrate (Fisher Scientific UK Ltd., Loughborough, UK; Catalogue No: S/4480/53)Sodium di-hydrogen orthophosphate dihydrate (Fisher Scientific UK Ltd., Loughborough, UK; Catalogue No: S/3760/53)2% Osmium tetroxide (TAAB Laboratories, Aldermaston, UK; Catalogue No. O015/1)Tannic acid, low molecular weight: FW 1000–1500 (Electron Microscopy Services, Hatfield, PA, USA; Catalogue No. 21710)Uranyl Acetate (TAAB Laboratories, Aldermaston, UK; Catalogue No. U007)Lead Acetate (Agar Scientific, Stansted, UK; Catalogue No: AGR1209)Absolute ethanol 99%+, extra pure SLR (Fisher Chemical E/0600DF/17)Acetone 99.5%, laboratory reagent (Fisher Scientific UK Ltd., Loughborough, UK; Catalogue No: 179973-1L)Araldite CY212 epoxy resin (Agar Scientific, Stansted, UK; Catalogue No: AGR1040)DDSA (Dodecenyl Succinic anhydride) resin hardener (Agar Scientific, Stansted, UK; Catalogue No: AGR1051)BDMA (Benzyl dimethylamine) Accelerator (Agar Scientific, Stansted, UK; Catalogue No: AGR1062)Stainless steel razor blades (Agar Scientific, Stansted, UK; Catalogue No: AGT586)7 mL glass specimen vials (Agar Scientific, Stansted, UK; Catalogue No: AGG284)Flat silicone embedding mould (Agar Scientific, Stansted, UK; Catalogue No: AGG3549)Silver epoxy conductive mounting medium (Agar Scientific, Stansted, UK; Catalogue No: AGG3349)Toluidine blue (Agar Scientific, Stansted, UK; Catalogue No: AGR1727)Di-sodium tetraborate (Merck KGAA, Darmstadt, Germany; Catalogue No. 106310)Luer-lock (optional) syringes 10 mL BD PlastiPak (Fisher Scientific UK Ltd., Loughborough, UK; Catalogue No: 15544835)0.2 µm Syringe filters, Sartorius Minisart™ RC (Fisher Scientific UK Ltd., Loughborough, UK; Catalogue No: 11740966)Plastic transfer pipettes, Fisherbrand™ 1 mL (Fisher Scientific UK Ltd., Loughborough, UK; Catalogue No: 13439118)Plastic transfer pipettes, Fisherbrand™ 3 mL (Fisher Scientific UK Ltd., Loughborough, UK; Catalogue No: 16405009)Glass microscope slides, menzel superfrost, 26 × 76 × 1 mm (Fisher Scientific UK Ltd., Loughborough, UK; Catalogue No: 16252171)Gatan 3View SEM aluminum pin stubs (TAAB Laboratories, Aldermaston, UK; Catalogue No. G312/1)

### 2.2. Equipment

pH Meter, Mettler Toledo™ S210 Seven Compact (Fisher Scientific UK Ltd., Loughborough, UK; Catalogue No: 15360161)Hot Plate Stirrer, Fisherbrand™ (Fisher Scientific UK Ltd., Loughborough, UK; Catalogue No: 15363518)Low speed Rotator (Agar Scientific, Stansted, UK; Catalogue No: AG1050)Vortex mixer (Starlab, Milton Keynes, UK; N2400-6110)Embedding oven, placed inside fume hood (Agar Scientific, Stansted, UK; Catalogue No: AGB7606)Diamond knife, Diatome, 3 mm 45 deg (TAAB Laboratories, Aldermaston, UK; Catalogue No. K065/30)UC6 ultramicrotome (Leica Microsystems (UK) Ltd., Milton Keynes, UK)ACE200 Low vacuum sputter coater (Leica Microsystems (UK) Ltd., Milton Keynes, UK)Zeiss Sigma Field Emission Gun Scanning Electron Microscope (Carl Zeiss, Oberkochen, Germany)Gatan 3view2XP (Gatan, Pleasanton, CA, USA)

## 3. Procedure

### 3.1. Fixation

First, prepare fresh 8% paraformaldehyde solution by adding 8 g paraformaldehyde powder to 100 mL distilled water in a conical flask. Depolymerise paraformaldehyde by heating to 65 °C on a hotplate in a fume hood, swirling the liquid occasionally. Add freshly made 1 M NaOH dropwise until the milky solution clears. Once cool, filter through a Whatman No. 1 filter paper. The stock solution can be stored at 4 °C for 1–2 weeks.

Prepare 0.2 M Sorenson’s sodium phosphate buffer by making up a solution (X) of 0.2 M dibasic sodium phosphate (Na_2_HP0_4_ 2H_2_O) by adding 3.56 g to 100 mL of distilled water. Then, prepare a solution (Y) of 0.2 M monobasic sodium phosphate (NaH_2_P0_4_2H_2_O) by adding 3.12 g to 100 mL of water. Mix 40.5 mL of (X) dibasic sodium phosphate with 9.5 mL of (Y) monobasic sodium phosphate and make up to 100 mL with distilled water and adjust pH, if necessary, to 7.4.

To prepare 100 mL of the working-strength Karnovsky fixative: 2% paraformaldehyde, 2.5% glutaraldehyde in 0.1 M phosphate buffer pH 7.4; add 25 mL of the 8% paraformaldehyde, 10 mL of stock, commercial 25% glutaraldehyde and 50 mL 0.2 M phosphate buffer together in a conical flask and mix the solution with a magnetic stirrer for 10 min. Check the pH is 7.4, adjust if necessary with 1 M HCl or 1 M NaOH, then make up the volume to 100 mL by adding up to 15 mL of distilled water, as required. Finally, filter the fixative through a Whatman number 1 filter paper.

Fix samples in 4 mL of the fixative per sample vial for 3 h at room temperature in cap-sealable, chemical-resistant 7 mL glass vials, then replace with 4 mL of 0.1 M Sorenson’s buffer.

**OPTIONAL STEP** Samples can be stored for up to six months in buffer at 4 °C. Buffer must be regularly exchanged for fresh every 2 weeks to prevent bacterial growth.


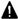
**CRITICAL STEP** Excise tissue from animals immediately after sacrifice and immerse in fixative to prevent autolysis of structures deep within the sample. Samples should be cut into blocks, 1–2 mm^3^ in size, to ensure the fixative penetrates fully and as quickly as possible.


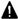
**CRITICAL STEP** Before starting the protocol, make up the following solutions in a fume hood the day before use. Prepare sufficient amounts of each for 4 mL per sample vial.

0.5% aqueous low molecular weight tannic acid solution. Weigh out tannic acid power, transfer into a 50 mL glass conical flask, add distilled water and stir gently until fully dissolved.Just prior to use, draw the solution up into a 20 mL Luer-lock syringe and attach a sartorius RC 0.20 µm syringe filter for dispensing into sample vials.2% aqueous uranyl acetate solution. Weigh powder directly into 50 mL polypropylene, solvent-resistant, universal tube with a sealable cap and add distilled water; double-wrap the cap with parafilm to prevent any leakage, and vortex until fully dissolved, then wrap the tube in aluminium foil as the uranyl acetate solution is light sensitive and store at room temperature.Just before use, draw the solution up into a 20 mL Luer-lock syringe and attach asartorius RC 0.20 µm syringe filter.1% ethanolic uranyl acetate solution. Weigh powder directly into a 50 mL solvent-resistant polypropylene universal tube and add ethanol. Double-wrap the sealed cap with parafilm to prevent leakage. Vortex the solution until the powder is dissolved, wrap the tube in aluminium foil and store at room temperature in the dark. Prior to use, draw the solution up into a 20 mL Luer-lock syringe and attach a sartorius RC 0.20 µm syringe filter.

### 3.2. En Bloc Staining

Use a rotator and plastic 1 mL Pasteur pipettes for all washing and staining steps.

Wash the samples by replacing the buffer with 4 mL of fresh 0.1 M Sorenson’s buffer three times, each for 10 min, to remove any traces of residual fixative solution. Then, replace with distilled water for 5 min. Wash samples for 30 min, with two 15-min changes of distilled water. During this washing step, prepare 1% aqueous osmium tetroxide solution from a commercial 5% solution supplied in 5 mL glass ampoules.

Add 4 mL per sample of 1% osmium tetroxide in distilled water for 1 h. After disposing of the osmium appropriately, wash samples with three 10-min changes of distilled water. Next, add 4 mL per sample of filtered 0.5% low molecular weight tannic solution for 2 h. Wash three times for 10 min with distilled water, then add 2% filtered aqueous uranyl acetate for 2 h.

**PAUSE STEP** The samples can be stored overnight in aqueous uranyl acetate in a fridge at 4 °C. Next, wash samples with three 10-min changes of distilled water.

### 3.3. Dehydration


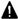
**CRITICAL STEP** Dehydration of the samples is crucial as any residual water would cause failure of resin polymerisation later in the protocol.

Dehydrate samples in the following sequential steps. Place samples in 70% ethanol for 20 min, 90% ethanol for 20 min, then two changes in 100% ethanol for 20 min each time.

Remove 100% ethanol from the samples and add the filtered 1% solution of uranyl acetate in ethanol for 2 h.


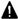
**CRITICAL STEP** The addition of 1% alcoholic UA for 2 h further enhances backscatter electron contrast of collagen fibrils improving overall imaging quality for SBF-SEM data acquisition. **Note:** Staining with 1% ethanolic UA must be kept to 2 h maximum. Prolonged staining can lead to excessive backscatter enhancement of collagen which may prevent visualisation of fibrillin fibres.


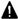
**CRITICAL STEP** During the 2-h *en bloc* uranyl acetate staining step, prepare lead acetate solution. The solution must be freshly made before use, otherwise the lead acetate will precipitate from solution within 3 h. Lead acetate solution is prepared by adding 1.4 g of lead acetate to a 50 mL plastic universal tube with sealable, screw-top plastic cap; 25 mL of 100% ethanol is added, the cap tightened and parafilm wrapped around the cap to further seal the tube ensuring no leakage. Vortex the solution, and gently shake the tube intermittently for 15 min. After 15 min, add 25 mL of 100% acetone to the ethanolic lead acetate solution. Reseal the cap with parafilm and vortex and shake the solution for a further 15 min. As the solute is in excess, it will appear as a milky white suspension with undissolved lead acetate present. Filter with Whatman 1 filter paper to remove the solid suspension and collect the clear solution into a clean 50 mL universal tube. Finally, draw up the lead solution into a 20 mL Luer-lock syringe and attach a 0.2 µm syringe filter.

After 2 h wash the sample with 100% ethanol two times for 10 min to remove excess uranyl acetate. Next, remove ethanol and add 1:1 100% ethanol and acetone solution for 10 min, followed by two further 15-min washes in the 1:1 solution. Then, add the lead acetate solution for 2 h. After lead *en bloc* staining, wash in 1:1 ethanol acetone solution two times for 15 min each.

The 1:1 ethanol acetone solution is then replaced with 100% acetone for 10 min followed by five further changes of acetone made every 10 min.


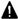
**CRITICAL STEP** The 100% acetone transition step is essential to remove any trace of ethanol, which is not miscible with the CY212 resin. Any residual ethanol would lead to unsuccessful polymerisation of resin blocks for SBF SEM 3view.

### 3.4. Resin Infiltration


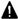
**CRITICAL STEP** Resin infiltration must be performed over a longer duration than normal TEM processing as the heavy metal solutions used in the previous steps make the sample extremely dense. To extend the duration of infiltration, the sample is exposed to the embedding resin in two stages: the first is without BDMA accelerator, and the second is with the complete resin mixture, including the accelerator. BDMA is used to catalyse the polymerisation reaction of the resin, so leaving out BDMA initially allows more time for infiltration.

Make up sufficient Araldite resin mixture for the whole embedding procedure, as a second batch may not mix consistently with one made earlier in which crosslinking will be already advanced. The full resin mixture contains the 3 components in the following proportions:
Araldite CY212 monomer14 mLDDSA Hardener16 mLBDMA Accelerator0.6 mL

Calculate the volume required, allowing approximately 3 mL per sample vial per change; as a rough guide, allow for 6 resin changes, both without and with BDMA accelerator, making a total of 12 exchanges of resin. This number is not absolute and can be reduced for very small specimens, or increased in the case of large or dense tissue samples, such as ligament or tendon. A small excess should also be added to accommodate the intermediate acetone: resin step (see below), and also for sufficient medium to fill wells in the embedding mould.

Make up the full volume required without BDMA accelerator in the first instance and split into two equal amounts, setting aside half of the resin mixture for the later addition of an appropriate volume of BDMA.

Prewarm the monomer and hardener by placing the reagent bottles in the embedding oven for 30 min at 60 °C to facilitate measuring and pouring. Additionally, prewarm a measuring cylinder and receiving glass conical flask, which will aid efficient dispensing and mixing of the two components. Both can be added to a suitable measuring cylinder, then poured into the warmed flask, which is swirled manually for several minutes to mix the contents thoroughly.

Allow air bubbles to rise to the surface and disperse before using.

Make up sufficient 1:1 mixture of 100% acetone and resin without BDMA to allow about 3 mL per specimen vial and transfer to vials for 1 h, with rotation.

Follow with two changes of the resin without accelerator at intervals of 1 h. Make one further change into resin without the accelerator and leave overnight on the rotator with vial caps in place.

On the following day, make 3 × 1–1.5-h changes with resin without BDMA, replacing vial caps after every exchange. Switch then to carry out 3 × 1–1.5-h changes with the full resin plus BDMA mixture. With the last resin change of the day, remove caps from the vials and leave on the rotator overnight.

On the subsequent day, make three further 2-h changes of full resin mixture before transferring the samples to blue silicon rubber EM moulds.


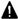
**CRITICAL STEP**. Embedding the specimens. Place the flask containing residual resin into the embedding oven for 10–15 min before inserting the specimens into wells in the mould. This will make the resin less viscous and assist in applying small amounts by pipette to accurately fill the mould wells. Ensure the samples are orientated in the mould so they can be cut in the correct section plane for 3view imaging. This can be facilitated by manipulating the specimens with fine forceps while observing via a dissecting microscope, placed temporarily into the fume hood. Small labels bearing brief printed details of the specimen can be immersed into the resin and positioned clear of the sample and prospective cutting face. Individual wells are then topped up with sufficient extra resin to form a level surface. Avoid over-filling the wells, as the convex surface so produced makes it difficult later to securely clamp the block in the specimen chuck of the ultramicrotome. The mould, complete with specimens, is then placed into an embedding oven contained within the fume hood. Polymerise the resin at 60 °C for 24 h minimum.

### 3.5. 3View Pin Preparation

Trim polymerised resin specimen blocks on an ultramicrotome to reveal the specimen within a 1.5 mm^3^ cube of resin. Cut the cube from the main block using a razorblade and attach to a Gatan 3view pin with super glue, such that the sectioning plane and region of interest (ROI) for SBF SEM image acquisition is facing upwards. Allow 30 min for the glue to set, then further secure the specimen block on the pin with silver epoxy conductive glue, being careful not to place glue on the surface for cutting. Leave specimen pins in a desiccator to dry thoroughly overnight. Using an ultramicrotome, secure the specimen pin in an ultramicrotome chuck and trim and polish the block face to create a smooth surface using a glass knife. Cut semi-thin (0.25–0.5 μm-thick) sections and transfer with a platinum loop to a droplet of distilled water on glass slides. Stain sections with 1% toluidine blue in 1% sodium tetraborate solution on a hot plate for 30 s, apply a coverslip and visualize using a light microscope to identify the ROI for SBF SEM data acquisition. Further trim the block face for a rectangular surface dimension of ~600 μm × 250 μm, or less containing the ROI. Sequence of pin preparation is shown in Figure 1.


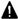
**CRITICAL STEP** The small block surface dimensions are required for serial block face cutting in the 3view 2XP system.

Sputter coat the prepared specimen pin with a diffuse layer of gold (8 nm thick), using a Leica ACE200 vacuum coating unit.


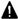
**CRITICAL STEP** Gold coating the specimen is critical for SBF SEM used in variable low-pressure mode, where excessive electrostatic charging from the poorly conductive epoxy resin block can interfere with imaging. The gold coating helps negate this charging effect by conducting the charge away from the block sides and block face.

## 4. 3View Imaging, Data Acquisition and 3D Reconstruction

Setting up the gold-coated sample pins for data acquisition and imaging follows standard operating procedures set out in the Gatan 3view 2XP operating manual.

### 4.1. Imaging and Data Acquisition

An SBF SEM system with a field emission source for high resolution imaging is required for the Imaging and 3D data acquisition.

In this protocol, a Zeiss Sigma FEG scanning electron microscope equipped with a Gatan 3view 2XP system was utilised.

For the expected outcome examples shown in Section 4, the following parameters were selected: 

Serial image acquisition set at 4 k × 4 k pixels.

Pixel resolution of either 4 or 8 nm and a pixel dwell time of 8 µs.

Accelerating voltage set at 3.4 kV with SEM vacuum set to a variable pressure mode of <28 Pa.

Data sets of 1000 images or more can be acquired through the automated cutting of the block surface using the Gatan 3view 2XP system. For elastic fibres, it is recommended that cutting thickness be set to between 50 to 100 nm to ensure accurate tracing of individual fibres through any serial image 3D data volumes generated.

**Note:** SEM imaging acquisition parameters may vary depending on the user’s imaging requirements.

### 4.2. Post-Data Processing

All raw serial image data generated from Gatan 3view system must follow a few post-data processing steps before importation into 3D visualisation software for reconstruction.


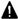
**CRITICAL STEP** The serial image data files collected in dm4 Gatan format must first be aligned using Gatan digital micrograph GMS3 software auto alignment tool.

Gatan dm4 aligned serial data files require conversion to a Tiff format to allow importation of the data into most 3D visualisation software packages.

### 4.3. 3D Reconstruction

The generation of 3D models can be undertaken using any 3D visualisation software packageinstalled on a graphics workstation. Some excellent open source software for 3D rendering of biological tissues is freely available on the web these include: IMOD https://bio3d.colorado.edu/imod/ (12 August 2021) and Microscope Imaging Browser http://mib.helsinki.fi/index.html (12 August 2021) specifically developed for SBF SEM. For the examples shown in this protocol, all 3D models were created using Amira 2020. 3.1 using semi-automated 3D rendering tools VOLREN and ISOSURFACE, which apply a sliding range (scalar fields) of differential light intensity of voxels to separate out different structures within a three-dimensional volume. The tools provide a quick, yet very effective way of reconstructing surface 3D models. This approach has been employed to illustrate how fibrillin-stained fibres and sheets can be easily rendered into 3D models using staining differential alone.

**Note** VOLREN and ISOSURFACE semi-automated rendering of surface structures will generate artefacts depending on the scalar field selected, which can appear as small speckles surrounding 3D semi-automated renderings. For accurate reconstructions of any biological structure using 3D visualisation software, manual tracing segmentation should always be the primary option.

## 5. Expected Results

The protocol for specimen processing gives high contrast to fibrillin-containing elastic fibers, when using scanning electron microscopy backscatter imaging, enabling segmentation and rendering into 3D reconstructions by image analysis software from large serial image datasets.

Three examples illustrate the results that can be achieved using the protocol:**Example A:** Demonstration of fibrillin elastic sheets 3D reconstructedfrom human cornea limbus ECM (Figure 2). Appendix A illustrates a fly-through of the 3D volume, which proceeds to reveal the intricate 3D surface model of the sheet created using a semi-automated segmentation approach.**Example B:** Demonstration of true elastic fibers 3D rendered from human cornea stroma ECM revealing their complex 3D organization and morphology (Figure 3). Appendix A illustrates the complex 3D arrangement of the elastic fibres within the corneal ECM.**Example C:** Demonstration of fine fibrillin fibers 3D rendered frommurineaort elastin sheet (Figure 4). Appendix A illustrates the fibrillin fibre arrangement within an elastin sheet in 3D.

All images, 3D models and Appendix A were created using Amira 3D visualization software.

## 6. Discussion: Including Limitations of the Protocol

It seems apparent that many of the most popular and effective tissue processing methods in current use for SBF SEM have arisen empirically from combinations of steps conventionally employed for contrasting ultrathin resin sections for transmission electron microscopy [17,22]. Early SBF SEM studies imaged membrane-rich neural tissues [12,13], contrasting intensely with reduced osmium tetroxide, sometimes enhanced by a second unreduced osmium step, with intermediate mordanting provided by thiocarbohydrazide or tannic acid solution. In this protocol, we describe a procedure by which elastic tissue components, located within the hypocellular, connective tissue matrices of the cornea and aorta can successfully be contrasted and identified with a single, unreduced osmium exposure. We use tannic acid as the mordant for subsequent heavy metal solutions, as previously described for dense collagenous tendon tissue [22,23].

Two previous SBF SEM studies of elastin in vascular tissues used single unreduced osmium tetroxide treatment. O’Connell et al. [24] included tannic acid within the primary glutaraldehyde fixative, followed by conventional osmium and uranyl acetate staining. Rezakhaniha et al. [25] used tannic acid as the mordant between unreduced osmium and palladium chloride. Both studies used relatively high concentrations of tannic acid (4–5%), which we suspect may increase the reported difficulty with penetration of this reagent, where 3 h minimum staining times and small specimens have been recommended [22]. This latter limitation would seem to be at odds with one of the major advantages of SBF SEM, namely, the potential for imaging large specimen volumes. Therefore, in view of these pitfalls, we recommend the use of the low molecular weight tannic acid (FW 1000–1500) employed here which, although not yet supported by comparative studies, appears to penetrate dense collagenous specimens effectively, and furthermore, as we show in images of corneal stroma, is able to enhance contrast in both amorphous elastin and fibrillin-rich, elastin-free microfibrils of the elastic fibre system.

Our choice of excess lead acetate in ethanol, first described as a stain for ultrathin sections by Kushida [21], for the final en bloc contrasting agent in this protocol is not based necessarily on superior staining over other lead salts, such as lead aspartate [17], which may well provide equally good contrast. Instead, its advantage lies in the fact that alcoholic lead is less liable to generate lead carbonate precipitates on air exposure [21], which are a common contaminant of specimens when aqueous lead salts are used. Exposure of samples to the lead solution is also carried out at room temperature, avoiding 60 °C incubation included in other protocols, which also increases the likelihood of lead precipitation and contamination of the specimen. Staining with lead in an ethanol–acetone vehicle, as we describe, is also readily compatible with subsequent transfer into epoxy resin, avoiding the need for toxic and inflammable intermediate reagents, such as propylene oxide.

The protocol can be used to investigate fibrillin ultrastructure in any type of biological specimen. Although the protocol provides enhanced staining of fibrillin, the stain also provides the same greyscale staining of other structures associated with standard SEM and TEM. Osmium used in the protocol works directly on unsaturated lipids in biological tissues and cells leading to the formation of mono-esters and dimeric mono-esters, which stain intensely black in backscattered SEM imaging. Therefore, highly osmiophilic, lipid-rich tissue could pose a problem when trying to discern the presence of dark intensely stained fibrillin-rich fibres. A simple work-around solution would be to remove osmium from the protocol. However, this could affect the overall integrity of cell and tissue ultrastructure which could be important when considering the context of fibrillin ultrastructural organization.

The protocol has been developed specifically for SBF SEM, but can be adapted for other vEM platforms, including Focused Ion Beam SEM (FIB SEM). The resin used in the protocol would then have to be exchanged for a harder resin, such as Durcupan, to permit ion beam milling of the specimen block.

## Figures and Tables

**Figure 1 mps-04-00056-f001:**
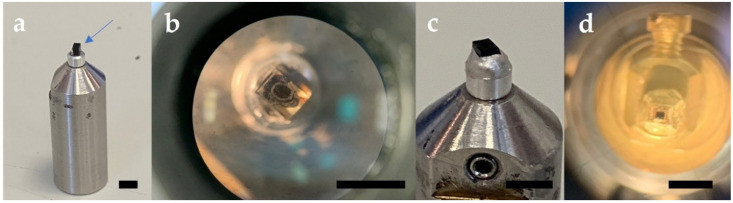
Pin preparation sequence (**a**) trimmed block containing specimen superglued to Gatan pin (blue arrow), (**b**) polished block face, (**c**) epoxy conductive glue applied around edges of block. (**d**) Specimen trimmed for cutting and gold coated. Scale bar black line 2.4 mm.

**Figure 2 mps-04-00056-f002:**
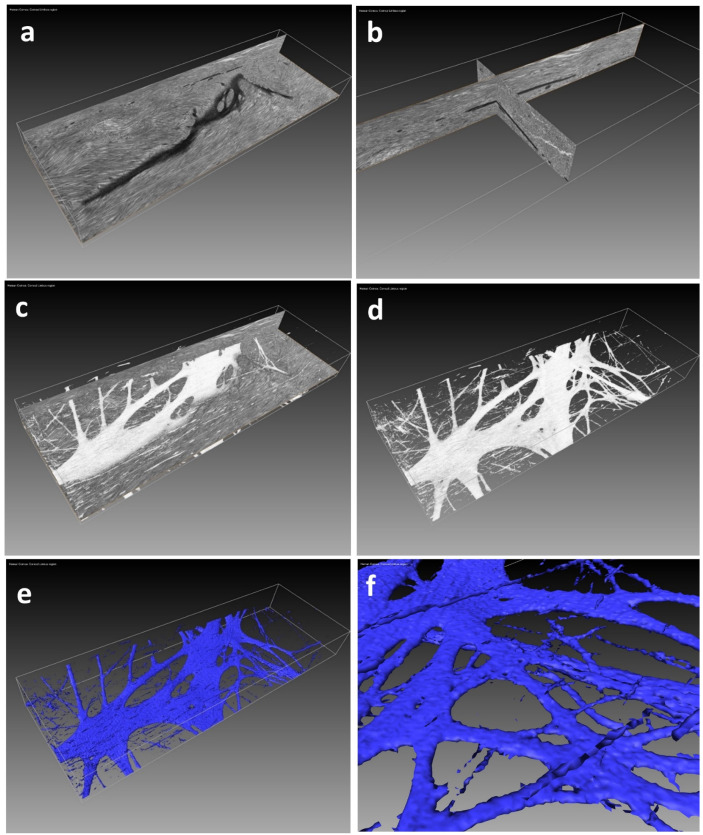
SBF SEM image data volume (45 × 20 × 10 microns) generated from human corneal limbus. The limbal region is characterized by an extracellular matrix (ECM) composed of hybrid type I and V collagen fibrils and sheets of fibrillin-rich elastic fibres. (**a**) demonstrates a dark intensely stained single fibrillin-rich elastic sheet, set within a light grey-stained ECM. (**b**) highlights how the dark intensely stained structure of the sheet is clearly seen within the x, y and z section planes of the 3D volume. (**c**) was obtained using the Amira Volren semi-automated rendering tool; the staining differential between the fibrillin-rich sheet and ECM is sufficient to render the sheet in 3D, shown in white. (**d**) reveals the 3D structure of the sheet with surrounding ECM removed. (**e**) was obtained utilizing Amira Isosurface; a solid surface (coloured blue) is applied to the sheet. (**f**) reveals the finer surface detail of the sheet.

**Figure 3 mps-04-00056-f003:**
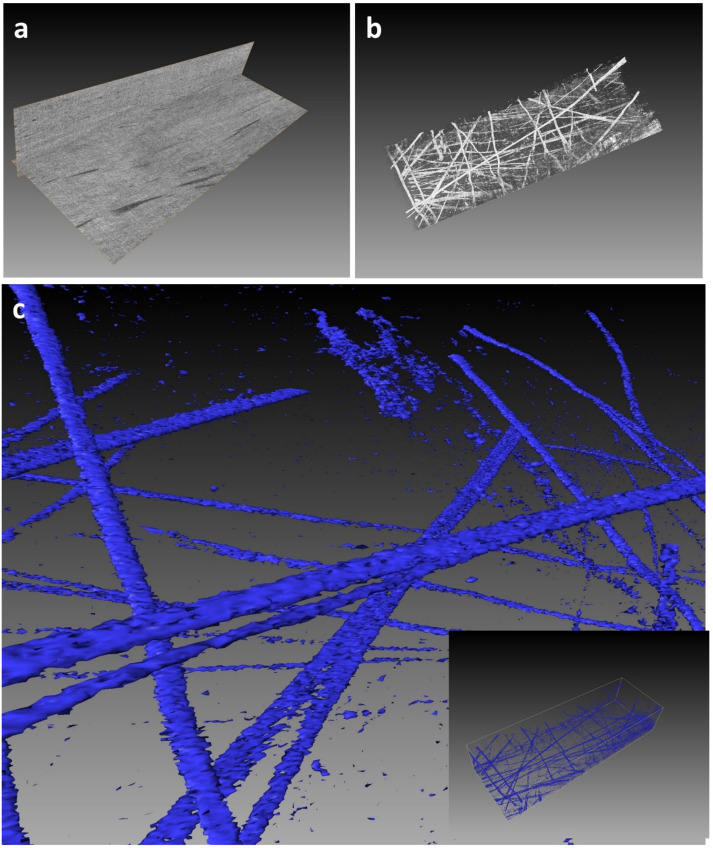
SBF SEM image data volume (50 × 25 × 15 microns) from human peripheral corneal stroma. The peripheral cornea stroma comprises an ECM of type I and V hybrid collagen fibrils and true elastic fibrillin-rich fibres [15]. (**a**) illustrates the presence of darkly stained elastic fibres within the volume. (**b**) Volren auto-rendering reveals the highly complex 3D arrangement of the fibres (white). (**c**) Iso-surface rendering of the elastic fibres (blue) highlights how the fine bifurcated, morphology of individual elastic fibres at nanometer resolution can be observed using the rendering tool.

**Figure 4 mps-04-00056-f004:**
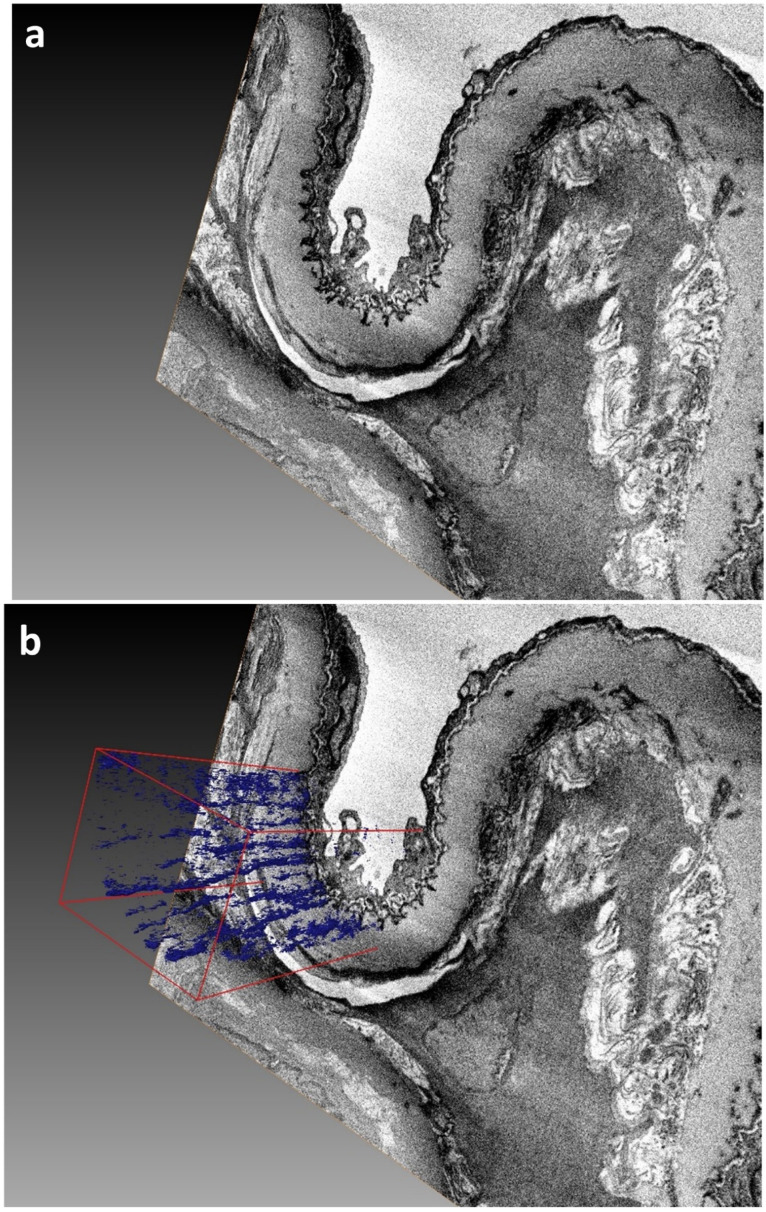
SBF SEM image data volume (40 × 25 × 25 microns) from a mouse aorta. The aorta is the main artery of the mammalian heart and in cross section comprises of numerous concentric layers of alternating collagenous ECM and elastin sheets, which surround a central lumen. (**a**) demonstrates an SBF SEM-acquired image of the aorta prior to 3D reconstruction. (**b**) reveals the arrangement of nanoscopic fibrillin micro-fibrils within one elastin sheet shown in blue using iso-surface rendering.

## Data Availability

Not applicable.

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
