# Peer review of "Contrast-Enhanced Tissue Processing of Fibrillin-Rich Elastic Fibres for 3D Visualization by Volume Scanning Electron Microscopy"

_mps, 2021, doi:10.3390/mps4030056_

Round 1
Reviewer 1 Report
Authors presented further development in a range of protocols aimed at serial block face SEM. These family of protocols relies on prolonged multistep contrasting procedure that involves series of electron dense stains. In short, authors introduce aqueous tannic acid, alcoholic uranyl acetate and alcoholic lead acetate to improve elastin contrast in tissues. Demonstrated image examples are of sufficient quality.
However, several things are important to mention:
- Tannic acid en bloc has been already shown to increase elastin contrast in tissues following the initial publication of Simionescu & Simionescu (1976). I was working with this reagent for quite a long time and did not have significant problem in stain penetration if tissue blocks were not too big.
- Tannic acid step for SBF SEM was already introduced 8 years ago to improve extracellular matrix contrast by Starborg et al. (2013). This publication is not mentioned whatsoever in the manuscript. And, probably, because of that there is no any discussion of why the presented advances are better than already published protocol.
- Also, lead salt contrasting step in the form of Walton’s lead aspartate is already present in original protocol by Deerinck et al.(2010). The manuscript does not explain why lead acetate stain is more preferable to lead aspartate one.
- It would be really beneficial for a reader to know what actual benefits these 3 modifications bring in comparison to already existing protocols. This manuscript does not include any comparison data or references of that kind.
- Without that comparison it is difficult to understand the purpose of each introduced step. For example, what are the benefits and potential problems of: a) adding alcoholic UA, b) using alcoholic lead acetate instead aqueous lead aspartate c) using osmium tetroxide instead of reduced osmium tetroxide as in Deerinck et al.(2010).
Summarizing all mentioned above. This manuscript has a great potential to move forward SBF SEM field in relation to elastin research. However, it is missing important pieces of information that can allow readers to assess the usefulness of proposed protocol. The manuscript may need some additional experimentation to be done to clarify these issues.
References:
Simionescu, N.; Simionescu, M.J. Galloylglucoses of low molecular weight as mordant in electron microscopy. II. The moiety
and functional groups possibly involved in the mordanting effect. Cell Biol. 1976 70, 622-33.
Starborg T, Kalson NS, Lu Y, et al. Using transmission electron microscopy and 3View to determine collagen fibril size and three-dimensional organization. Nat Protoc. 2013;8(7):1433-1448. doi:10.1038/nprot.2013.086
Deerinck, T.J.; Bushong, E.A.; Lev-Ram, V.; Shu, X.; Tsien, R.Y.; Ellisman, M.H. Enhancing Serial Block-Face Scanning Electron Microscopy to Enable High Resolution 3-D Nanohistology of Cells and Tissues. Microsc Microanal 2010 16, 1138-1139.
Reviewer 2 Report
The manuscript is well written, and each step of the protocol is clearly explained. The following minor points are to be addressed by authors before this paper is accepted for publication.
- In the Introduction section, the authors should also mention the shortcomings of traditional histological stains. Specifically, it would be good to highlight the utility of the protocol used by the researchers as compared to other approaches.
- The paraformaldehyde pH should be adjusted to 7.4 after it dissolves.
- It will be good to provide the protocol for phosphate preparation.
- In example B, figure 3c, the morphology of elastic fibers at nanometer resolution can be observed. However, a lot of background noise can also be seen. Removal or reductions of background will be better.
- Some typographical errors:
- Line 170: To prepare 100 ml of the working-strength Karnovsky fixative: 2 % paraformaldehyde, 2.5 % glutaraldehyde in 0.1 M buffer pH 7.4; - correct it to phosphate buffer.
- Line 180: 2 % aqueous uranyl acetate solution. Weigh power directly into 50 ml polypropylene – correct it to powder.
- Line 321: Accelerating voltage set at 3.4 Kv with SEM vacuum- correct it to kV.
Round 2
Reviewer 1 Report
I thank the authors for introduced amendments that improved the internal logic significantly. I would recommend to publish the manuscript is its present form.